# Trends in extent of surgical cytoreduction for patients with ovarian cancer

**Deanna H. Wong**[1], **Alexandra L. Mardock**[1], **Erica N. Manrriquez**[2], **Tiffany S. Lai**[2], **Yas Sanaiha**[3], **Abdulrahman K. Sinno**[4], **Peyman Benharash**[3], **Joshua G. Cohen**[2]*

1 David Geffen School of Medicine, University of California, Los Angeles, Los Angeles, California, United States of America, 2 Division of Gynecologic Oncology, Department of Obstetrics and Gynecology, University of California, Los Angeles, Los Angeles, California, United States of America, 3 Division of Cardiac Surgery, Cardiovascular Outcomes Research Laboratories (CORELAB), David Geffen School of Medicine, University of California, Los Angeles, Los Angeles, California, United States of America, 4 Division of Gynecologic Oncology, Department of Obstetrics and Gynecology, University of Miami, Florida, United States of America

* JGCohen@mednet.ucla.edu

**Data Availability Statement:** All relevant data are within the manuscript and its Supporting information files.

## Abstract

### Purpose

To identify patient and hospital characteristics associated with extended surgical cytoreduction in the treatment of ovarian cancer.

### Methods

A retrospective analysis using the National Inpatient Sample (NIS) database identified women hospitalized for surgery to remove an ovarian malignancy between 2013 and 2017. Extended cytoreduction (ECR) was defined as surgery involving the bowel, liver, diaphragm, bladder, stomach, or spleen. Chi-square and logistic regression were used to analyze patient and hospital demographics related to ECR, and trends were assessed using the Cochran-Armitage test.

### Results

Of the estimated 79,400 patients undergoing ovarian cancer surgery, 22% received ECR. Decreased adjusted odds of ECR were found in patients with lower Elixhauser Comorbidity Index (ECI) scores (OR 0.61, p<0.001 for ECI 2, versus ECI≥3) or residence outside the top income quartile (OR 0.71, p<0.001 for Q1, versus Q4), and increased odds were seen at hospitals with high ovarian cancer surgical volume (OR 1.25, p<0.001, versus low volume). From 2013 to 2017, there was a decrease in the proportion of cases with extended procedures (19% to 15%, p<0.001). There were significant decreases in the proportion of cases with small bowel, colon, and rectosigmoid resections (p<0.001). Patients who underwent ECR were more likely treated at a high surgical volume hospital (37% vs 31%, p<0.001) over the study period. For their hospital admission, patients who underwent ECR had increased mortality (1.6% vs. 0.5%, p<0.001), length of stay (9.6 days vs. 5.2 days, p<0.001), and mean cost ($32,132 vs. $17,363, p<0.001).

**Funding:** DHW received funding by the Dean's Leadership in Health and Science Scholarship at the David Geffen School of Medicine. The funders had no role in study design, data collection and analysis, decision to publish, or preparation of the manuscript.

**Competing interests:** The authors have declared that no competing interests exist.

## Conclusions

Likelihood of ECR was associated with increased medical comorbidity complexity, higher income, and undergoing the procedure at high surgical volume hospitals. The proportion of ovarian cancer cases with ECR has decreased from 2013–17, with more cases performed at high surgical volume hospitals.

## Introduction

Approximately 22,000 women in the United States each year are diagnosed with ovarian cancer, which continues to be the fifth most common cause of cancer-related death among women [1, 2]. Five-year survival across all women diagnosed with ovarian cancer is 48% [1], but each patient's prognosis is influenced by multiple variables including stage, social determinants of health, and volume of residual disease after surgical cytoreduction [3].

Due to screening and diagnostic limitations, 80% of patients are diagnosed with stage III or IV cancer [3]. At these advanced stages, survival is significantly decreased and treatment is more challenging due to spread of disease beyond the ovary. One important predictor of progression-free and overall survival is complete cytoreduction to no gross residual disease [4, 5], which is a practice recommended by the National Comprehensive Cancer Center (NCCN) [6] and is an important modifiable prognostic factor for ovarian cancer. Extended cytoreduction (ECR) is often required to achieve this state and involves surgical procedures beyond the standard hysterectomy and bilateral salpingo-oophorectomy. These additional procedures include tumor resection involving the bowel, diaphragm, liver, spleen, bladder, or stomach.

Receiving ECR and reaching complete cytoreduction have been shown to correlate with surgeon case volume [7, 8]. Studies show that ovarian cancer patients treated in the United States have improved surgical outcomes and survival when treated in higher volume hospitals and by gynecologic oncologists [7, 8]. However, not all patients have equal access to these resources. Age, race, income, and geography may limit a patient from being treated at higher volume hospitals, by gynecologic oncologists, and in accordance with NCCN guidelines [3, 9–13].

The objective of this study was to identify patient and hospital characteristics associated with ECR and to identify trends in extended procedures over the study time period. We hypothesized that patients with higher income or private insurance, those seen at hospitals with higher procedural volume, and those with greater disease burden would have increased likelihood of receiving ECR. Additionally, we postulated that ECR would be associated with higher risk of perioperative mortality and longer, more costly hospitalization.

## Methods

### Study design and data source

We performed a retrospective analysis of patient hospitalizations between January 1, 2013 and December 31, 2017 as identified by the National Inpatient Sample (NIS). The NIS is the largest publicly available all-payer inpatient database in the United States and is part of the Agency for Healthcare Research and Quality's (AHRQ) Healthcare Cost and Utilization Project. The database contains data from more than 7 million annual hospital discharges in 48 states and, through inherent survey weights, estimates over 35 million hospitalizations annually [14]. This

study was granted exemption from the Institutional Review Board at the University of California, Los Angeles given its use of deidentified data.

## Population and surgical procedures

The study cohort consisted of female patients aged 18 years or older who underwent any surgery involving oophorectomy and carried a diagnosis of ovarian, fallopian tube, or primary peritoneal cancer, as indicated by the *International Classification of Disease*–Ninth and Tenth Revisions–Clinical Modification (ICD-9 and ICD-10). Secondary procedure codes were used to identify patients who had undergone ECR, which was defined as any additional surgical procedure involving ileostomy/colostomy placement, splenectomy, bladder resection or resection of the colon, small intestine, liver, diaphragm, or stomach (Supplement 1) [15]. Given the national transition from ICD-9 to ICD-10 coding in October 2015, accurate conversion of procedure codes was confirmed using chi-square tests comparing the third and fourth quarters of 2015. This was performed for the proportion of each unique procedure that was studied in trend analysis—total extended procedures, small bowel resection, colon resection, rectosigmoid resection, and colostomy formation—and no significant change was found in the proportions identified by ICD-9 versus ICD-10 codes.

## Baseline patient and hospital characteristics

Patient demographics, clinicopathologic information, and treatment outcomes were obtained from the database, as were baseline characteristics of all treating institutions. Variables included patient age, race, primary payer, and income quartile by residential ZIP code. Comorbidities were identified and summarized using the Elixhauser Comorbidity Index (ECI), a validated tool based on 30 comorbidities identified using ICD-9 and ICD-10 codes [16].

Hospital characteristics included bed size, teaching status, geographic region, and ovarian cancer surgical volume. Hospital surgical volume was categorized as low, medium, or high based on ovarian cancer cytoreduction volume per institution per year, divided into tertiles. This was accomplished using the unique hospital identification numbers assigned each year within the NIS. Mortality was defined as death during the hospital admission.

## Statistical analysis

The primary outcome of this study was utilization of ECR across the study period. Secondary outcomes included factors independently associated with ECR, trends of extended procedure including hospital surgical volume over time, inpatient mortality, hospitalization cost, and length of stay in relation to ECR.

All numerical observations listed in this study were generated using survey weights included in the NIS database to provide nationally-representative estimates. Patient demographics among those undergoing extended versus non-extended procedures were compared using chi-square univariate analysis for categorical variables and adjusted Wald tests for continuous variables.

Multivariable logistic regression models were used to estimate the impact of the primary predictors (age, race, insurance status, income quartile by ZIP code, hospital geographic region, and hospital surgical volume) on the likelihood of receiving extended cytoreduction. The reference group was set as the majority group within each category. Trend analysis across the study period was conducted using the Cochrane-Armitage test for trend of proportions. All analysis was performed using Stata Version 15.1 (Stata Corporation, College Station, TX), with statistical significance set at $p < 0.05$.

## Results

We identified 79,400 women who were admitted and underwent surgery for ovarian, fallopian tube or primary peritoneal cancer between January 1, 2013 and December 31, 2017. Of these patients, 22% underwent ECR. Patients who underwent extended cytoreductive procedures were primarily aged 60–69, more likely to have an ECI $\geq$3 (77% vs 59%, p<0.001), insured through Medicare (46% vs. 38%, p<0.001), residing in ZIP codes with the highest median income (31% vs. 27%, p<0.001), and were more likely treated at a high surgical volume hospital (37% vs 31%, p<0.001). For their hospitalization, these patients had an increased mortality (1.6% vs. 0.5%, p<0.001), length of stay (9.6 days vs. 5.2 days, p<0.001), and mean cost ($32,132 vs. $17,363, p<0.001). Demographics are listed in Table 1.

To determine whether surgical volume was associated with ECR, multivariate logistic regression was performed adjusting for age, race, insurance status, and income quartile. Surgical volume was an independent predictor of receiving ECR. Decreased odds of receiving ECR were found in patients with ECI of <3 (OR 0.21 [0.18–0.26] and OR 0.61 [0.55–0.67] for ECI 1 and 2, respectively) and those residing in ZIP codes outside the top income quartile (OR 0.71 [0.63–0.81], p<0.001 for Q1, versus Q4). Detailed results are listed in Table 2.

To further characterize the types of extended procedures performed, the annual percentages of each component of extended procedure were calculated. On average, rectosigmoid resection was performed in 10.8%, colon resection in 10.5%, small bowel resection in 3.7%, splenectomy in 3.0%, diaphragm resection in 2.7%, colostomy in 2.0%, ileostomy in 1.8%, bladder resection in 1.1%, hepatic resection in 1.0%, and gastrectomy in 0.9% of all cytoreduction cases.

In assessing the trends of ECR over the study period, the annual proportion of ECR performed each year decreased significantly from 19% to 15% over the study period (Fig 1). There was a significant decrease in the annual proportion of ECR procedures including rectosigmoid resections (p<0.001), colon resections (p<0.001), small bowel resections (p = 0.001), and colostomies (p<0.001). The trends over time per procedure are shown in Fig 2.

The proportion of extended procedures performed at low, medium, and high-volume hospitals changed over time significantly. By the end of the five-year study period, there was an increase in the proportion of cases at high-volume hospitals from 38.4% to 42.9%, (p = 0.001) and a decrease in the proportion at medium-volume hospitals from 24.7% to 19.1% (p = 0.011) (Fig 3).

Compared with the rest of the study population, patients undergoing ECR had higher postoperative mortality during their hospital admission (1.6% vs. 0.5%, p<0.001), longer length of index hospitalization (9.6 vs. 5.2 days, p<0.001), and a higher total mean cost of care ($32,132 vs. $17,363, p<0.001) for this hospitalization.

## Discussion

Standard ovarian cancer treatment typically requires a multimodal approach including both surgery and chemotherapy. Improved survival outcomes are associated with optimal surgical cytoreduction to no gross residual disease [4, 5]. In order to achieve this, extended cytoreductive procedures involving the bladder, spleen, bowel, or other abdominal organs are sometimes necessary. Our data suggest an association between ECR, hospital surgical volume, and certain patient demographic factors. While ECR has decreased over the study period, patients undergoing ECR experience a higher rate of perioperative mortality with longer hospitalization and cost of care compared to other ovarian cancer patients.

Our data indicate that ECR in ovarian cancer surgery is associated with certain patient demographics. Women with more comorbidities as identified with an ECI of 3 or more were more likely to undergo ECR. This may be attributed to late presentation to care or other factors

**Table 1. Population demographics stratified by occurrence of extended cytoreductive procedure.**

| | Not extended (N = 61,850, 77.9%) | Extended (N = 17,550, 22.1%) | *P*-value |
|---|---|---|---|
| **Age** | | | |
| <40 years | 4570 (7.4%) | 635 (3.6%) | <0.001 |
| 40–49 years | 8145 (13.2%) | 1585 (9.0%) | <0.001 |
| 50–59 years | 15720 (25.4%) | 4000 (22.8%) | 0.0013 |
| 60–69 years | 17680 (28.6%) | 5730 (32.6%) | <0.001 |
| 70–79 years | 11730 (19.0%) | 4345 (24.8%) | <0.001 |
| 80+ years | 4005 (6.5%) | 1255 (7.2%) | 0.1579 |
| **Year** | | | |
| 2013 | 12220 (19.8%) | 3935 (22.4%) | 0.0062 |
| 2014 | 11955 (19.3%) | 3825 (21.8%) | 0.0104 |
| 2015 | 12150 (19.6%) | 3625 (20.7%) | 0.2973 |
| 2016 | 12805 (20.7%) | 3070 (17.5%) | 0.0019 |
| 2017 | 12720 (20.6%) | 3095 (17.6%) | 0.0018 |
| **Race** | | | |
| White | 43690 (70.6%) | 12845 (73.2%) | 0.0053 |
| Black | 4685 (7.6%) | 1260 (7.2%) | 0.4454 |
| Hispanic | 5115 (8.3%) | 1185 (6.8%) | 0.003 |
| API | 2605 (4.2%) | 705 (4.0%) | 0.6175 |
| Other / Unknown | 5755 (9.3%) | 1555 (8.9%) | 0.466 |
| **Payer** | | | |
| Private | 28150 (45.5%) | 7340 (41.8%) | 0.0002 |
| Medicaid | 6105 (9.9%) | 1320 (7.5%) | <0.001 |
| Medicare | 23760 (38.4%) | 8050 (45.9%) | <0.001 |
| Other or unknown | 3835 (6.2%) | 840 (4.8%) | 0.0018 |
| **Median Income by ZIP Code** | | | |
| Q1 (lowest) | 13415 (21.7%) | 3420 (19.5%) | 0.0088 |
| Q2 | 15335 (24.8%) | 3910 (22.3%) | 0.0028 |
| Q3 | 16110 (26.0%) | 4385 (25.0%) | 0.2041 |
| Q4 | 15870 (25.7%) | 5450 (31.1%) | <0.001 |
| **Elixhauser comorbidity sum** | | | |
| 1 | 10055 (16.3%) | 730 (4.2%) | <0.001 |
| 2 | 15070 (24.4%) | 3245 (18.5%) | <0.001 |
| 3+ | 36725 (59.4%) | 13575 (77.4%) | <0.001 |
| **Hospital Bed Size** | | | |
| Small | 5240 (8.5%) | 1505 (8.6%) | 0.8771 |
| Medium | 14355 (23.2%) | 3530 (20.1%) | 0.001 |
| Large | 42255 (68.3%) | 12515 (71.3%) | 0.0054 |
| **Hospital Teaching Status** | | | |
| Nonteaching | 8225 (13.3%) | 2335 (13.3%) | 0.9927 |
| Teaching | 53625 (86.7%) | 15215 (86.7%) | 0.9927 |
| **Hospital Region** | | | |
| Northeast | 12645 (20.4%) | 3360 (19.1%) | 0.231 |
| Midwest | 13845 (22.4%) | 4085 (23.3%) | 0.3573 |
| South | 21045 (34.0%) | 5915 (33.7%) | 0.7761 |
| West | 14315 (23.1%) | 4190 (23.9%) | 0.4494 |
| **Hospital Ovarian Cancer Surgical Volume** | | | |
| Low | 26095 (42.2%) | 6545 (37.3%) | <0.001 |

(*Continued*)

**Table 1.** (Continued)

| | Not extended (N = 61,850, 77.9%) | Extended (N = 17,550, 22.1%) | *P*-value |
|---|---|---|---|
| Medium | 16340 (26.4%) | 4475 (25.5%) | 0.367 |
| High | 19415 (31.4%) | 6530 (37.2%) | <0.001 |
| Mortality and Resource Utilization | | | |
| Mortality | 295 (0.5%) | 285 (1.6%) | <0.001 |
| Mean Length of Stay (days) [95% CI] | 5.2 [5.1–5.3] | 9.6 [9.3–9.9] | <0.001 |
| Mean cost (USD) [95% CI] | 17,363 [17,053–17,673] | 32132 [30,940–33,325] | <0.001 |

Values reported as N (%) or mean [95% CI].

API, Asian/Pacific Islander; CI, confidence interval; USD, United States dollars.

contributing to disease burden. Balancing the survival benefits of lower residual disease to the higher postoperative morbidity and mortality associated with ECR, risk stratification should be used for optimal treatment planning [17, 18]. Emphasis on preoperative medical optimization and focus on shared decision making with patients may allow for improved perioperative outcomes in patients with numerous medical co-morbidities.

Socioeconomic factors and insurance status have been associated with access and type of surgical treatment. We found that residing in the highest income ZIP codes was an independent predictor of ECR (p<0.001), which is consistent with findings from previous studies [19]. Other studies have found that having private insurance was associated with better health-related quality of life; however, our findings suggest that private insurance does not influence odds of extended cytoreductive surgery in ovarian cancer [11, 20, 21]. High quality care, including surgical management, should be the goal for every patient. Further understanding of the impact social determinants of health have in the care of women with ovarian, fallopian tube, or primary peritoneal cancer will allow for improved perioperative management strategies and equal access to care.

In this report, the proportion of cases with extended procedures decreased from 2013–2017. This finding goes beyond the years studied by Jones et al., who demonstrated that from 1998 to 2013, ECR had increased from 1998 to 2010, declined in 2011, then rose again from 2012 to 2013 [15]. The use of neoadjuvant chemotherapy has been associated with increased likelihood of complete cytoreduction. With the increased use of neoadjuvant chemotherapy from 30% in 2010 to 39% in 2016, it is likely there is a decreased need for extended procedures during cytoreduction [22–25].

Cases were found to be increasingly performed at high volume centers, which may indicate patients are being referred to tertiary centers for specialized care. As shown in this study, patients undergoing primary debulking surgery at high volume centers have also been previously shown to be an independent predictor of ECR [26]. In addition, high volume centers have been associated with a higher likelihood of achieving complete gross resection compared to lower volume centers [27]. Surgical care of women with ovarian cancer has been trending towards centralized care, concentrating on the number of surgeons and hospitals, which has been associated with decreased perioperative mortality [28].

The decrease in the proportion of patients undergoing bowel resections seen over the study period is likely of multifactorial cause. One potential contributing cause is the increasing use of neoadjuvant chemotherapy, which may decrease tumor burden and reduce the need for extended procedures. This highlights the importance for standardization of care, with

**Table 2. Independent predictors of receiving ECR.**

|  | Odds Ratio | [95% CI] | P-value |
|---|---|---|---|
| Age |  |  |  |
| <40 | 0.60 | [0.48–0.74] | <0.001 |
| 40–49 | 0.71 | [0.62–0.83] | <0.001 |
| 50–59 | 0.84 | [0.75–0.94] | 0.003 |
| 60–69 | [REF] |  |  |
| 70–79 | 1.09 | [0.97–1.22] | 0.151 |
| 80+ | 0.90 | [0.76–1.07] | 0.243 |
| Elixhauser Comorbidity Index |  |  |  |
| 1 | 0.21 | [0.18–0.26] | <0.001 |
| 2 | 0.61 | [0.55–0.67] | <0.001 |
| 3+ | [REF] |  |  |
| Race |  |  |  |
| White | [REF] |  |  |
| Black | 0.98 | [0.84–1.15] | 0.844 |
| Hispanic | 0.95 | [0.81–1.11] | 0.504 |
| API | 1.03 | [0.84–1.26] | 0.787 |
| Other or unknown | 0.94 | [0.81–1.09] | 0.413 |
| Payer |  |  |  |
| Private | [REF] |  |  |
| Medicaid | 0.88 | [0.76–1.03] | 0.119 |
| Medicare | 0.92 | [0.81–1.03] | 0.137 |
| Other or unknown | 0.90 | [0.75–1.09] | 0.294 |
| Income Quartile of Residential ZIP Code |  |  |  |
| <25th percentile | 0.71 | [0.63–0.81] | <0.001 |
| 25th-50th percentile | 0.72 | [0.64–0.81] | <0.001 |
| 50th-75th percentile | 0.78 | [0.70–0.87] | <0.001 |
| >75th percentile | [REF] |  |  |
| Hospital Region |  |  |  |
| Northeast | 0.83 | [0.72–0.96] | 0.013 |
| Midwest | 0.99 | [0.87–1.12] | 0.847 |
| South | [REF] |  |  |
| West | 1.02 | [0.90–1.16] | 0.785 |
| Hospital Volume |  |  |  |
| Low | [REF] |  |  |
| Medium | 1.04 | [0.93–1.17] | <0.001 |
| High | 1.25 | [1.12–1.40] | <0.001 |

API, Asian/Pacific Islander; CI, confidence interval; REF, reference group.

prospective data collection of outcomes. With the increasing use of neoadjuvant chemotherapy and trend in decreased ECR, measured surgical quality metrics will allow for gynecologic oncologists to maintain consistent levels of cytoreductive care whether patients are undergoing upfront or interval cytoreductive surgery.

Patients receiving ECR were associated with increased perioperative mortality, which likely reflects the greater disease burden requiring extended debulking procedures, as well as medical co-morbidities and the complexities of surgery. ECR is associated with a postoperative mortality of approximately 1.8%, which corresponds with our findings [29, 30]. Similarly, the

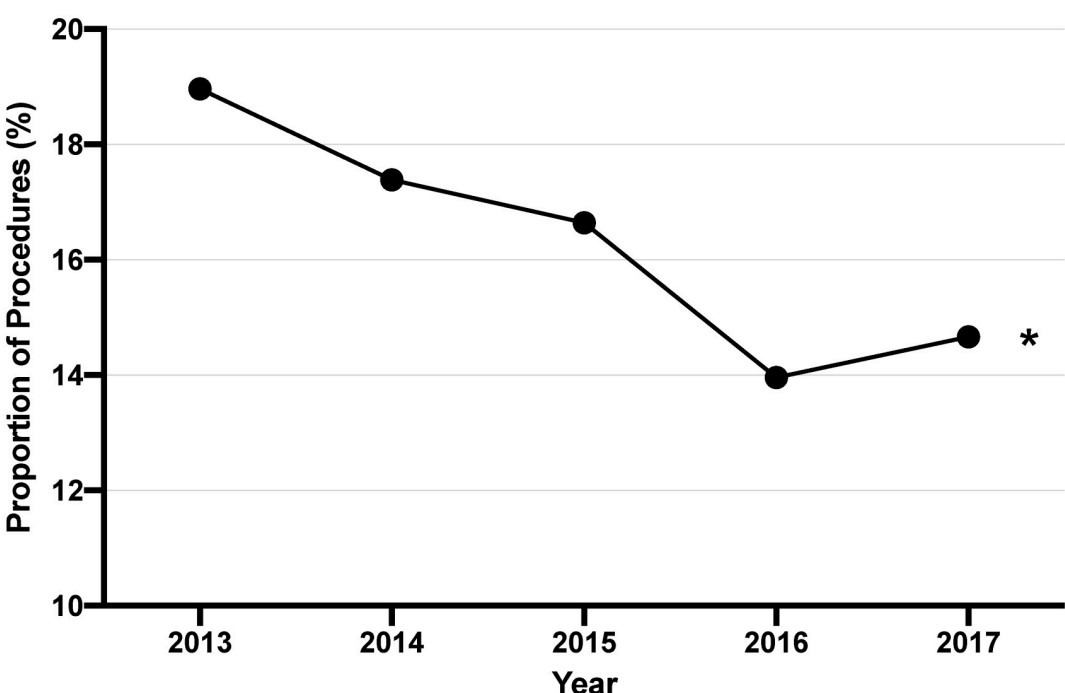

**Fig 1. The proportion of extended procedures across all hospitals each year.** *Significance at $P<0.05$.

increased length of stay and corresponding higher cost of hospitalization for patients with ECR has been associated with the dissemination of disease, extent of surgery, and presence of comorbidities, consistent with our findings [31]. Gynecologic oncologists and stakeholders in the management of women with ovarian cancer must work to establish guidelines that require adherence and quality metrics that are measured in a national database similar to the National Surgical Quality Improvement Project (NSQIP). NSQIP has been used to improve the care of patients with colon and rectal cancer in a similar fashion [32]. Prospective data collection will allow for a better understanding of the role of surgical volume and patient demographics associated with ovarian cancer outcomes. This will be more important now than in the past as ECR procedures become less common with the use of neoadjuvant chemotherapy and current trainees may not have as much experience with radical tumor debulking prior to entering practice.

Strengths of this study include its use of a nationally-representative database with a large sample size and detailed information regarding age and socioeconomic status. Limitations include the retrospective study design and the dataset used. The identification of patients undergoing ECR within the NIS database was dependent upon billing codes, the accuracy of which may vary between institutions including the completeness of the coding. The database also does not allow for tracking of patients across multiple hospitalizations, so we are unable to determine the number of new ovarian cancer cases per year as it relates to our trend analysis. In addition, these codes do not report surgeon-specific data on cytoreduction, nor do they differentiate whether surgery was in the setting of upfront or neoadjuvant therapy. Nevertheless, timing of surgery should not impact the findings of this study which was to determine patient and hospital factors associated with undergoing ECR. Although the NIS includes information on a variety of health conditions, it lacks granular information specific to ovarian cancer, such as disease stage and histologic subtype. Finally, the database does not include information on

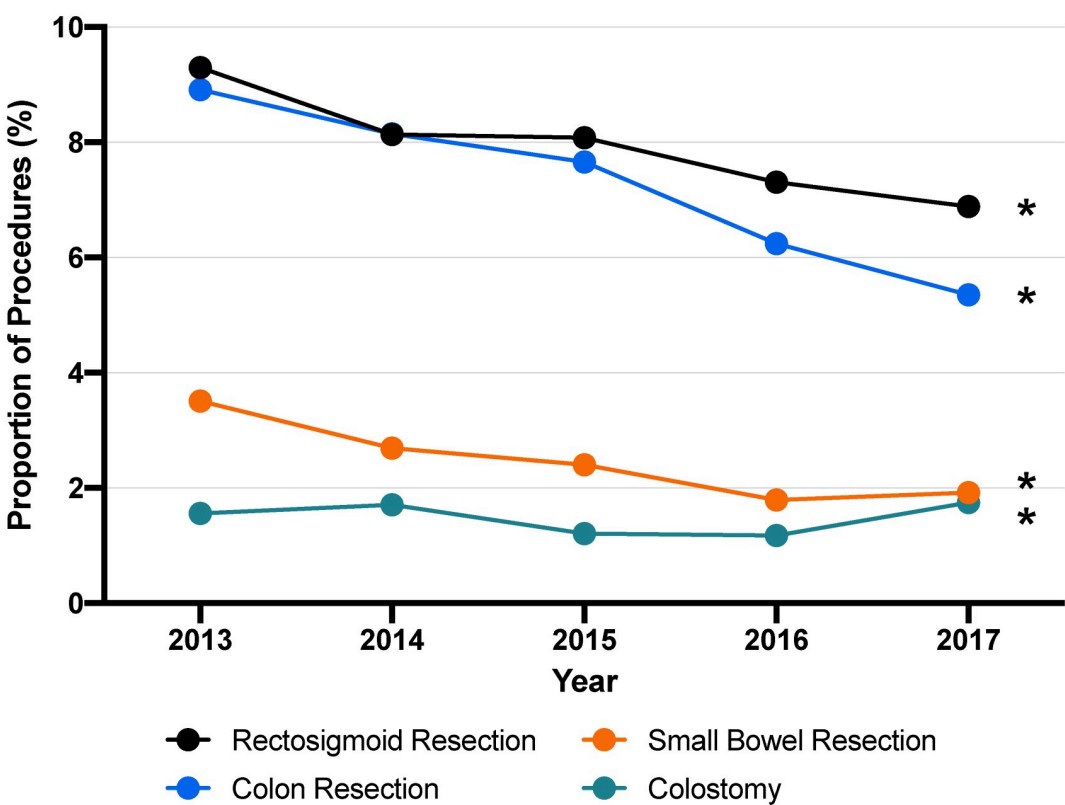

**Fig 2. The proportion of all cytoreduction procedures involving the gastrointestinal tract each year.** *Significance at $P<0.05$.

the amount of residual disease after surgery. While lack of this data may be limiting, physicians frequently underestimate the rate of optimal cytoreduction, with one study reporting 40% of cases were inaccurately described as optimal [33]. Thus, since previous studies have shown that ECR increases the odds of reducing to minimal residual disease [34, 35], we focused on rates of ECR as a proxy for improved odds of optimal cytoreduction.

Our data highlights the current trends in ECR for patients with ovarian cancer. There is a decreasing trend in ECR, however these procedures continued to be associated with increased perioperative mortality, length of stay, and cost. ECR is a mainstay in the management of ovarian cancer and should remain a priority in gynecologic oncology training. A large-scale effort by stakeholders in the treatment of women with ovarian cancer must take place to ensure adequate care for these patients irrespective of socioeconomic status and hospital volume. Similar to what has been done in other fields such as rectal cancer, an ongoing database with metric outcomes is likely warranted. Future studies would benefit from use of institutional or ovarian cancer-specific databases that may better characterize the quality of ECR and its relation to patient centered outcomes.

In conclusion, our findings highlight an association between undergoing extended surgical cytoreduction, socioeconomic factors, and surgical volume of the treating institution. Over time, the proportion of cases with ECR has been decreasing, and more of these cases are being performed at high volume centers. Extended procedures performed are decreasing proportionally each year, which has important implications for the quality of care provided to ovarian cancer patients.

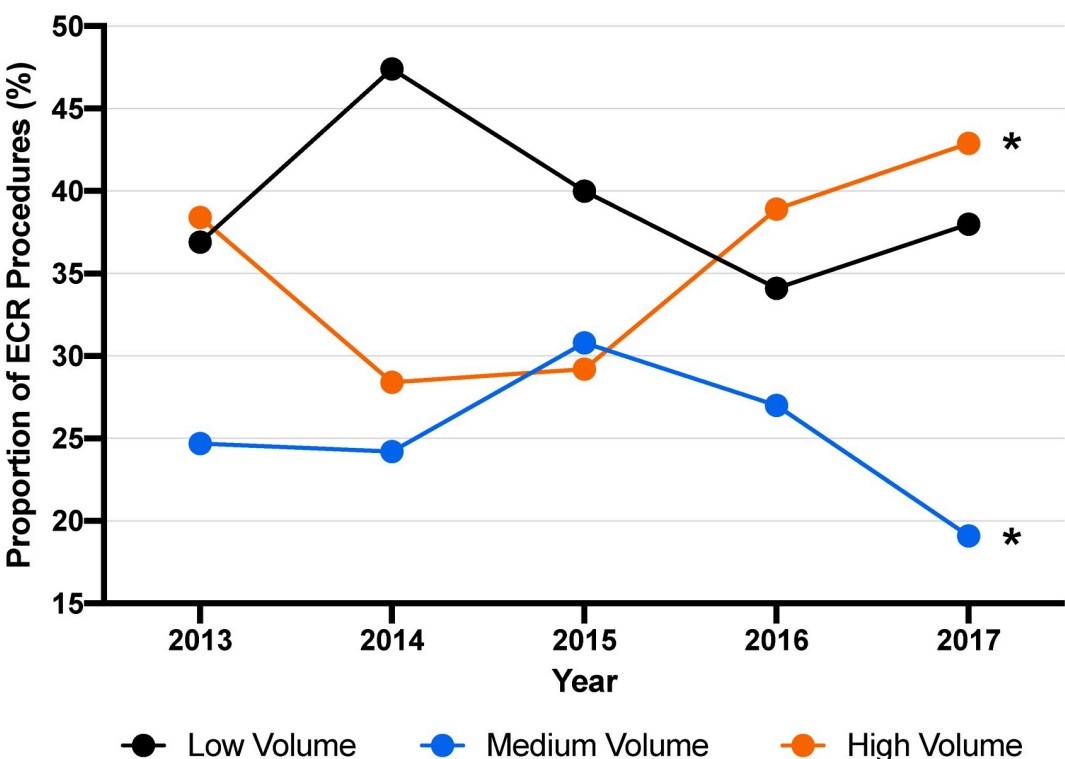

**Fig 3. The proportion of extended procedures performed at high, medium, and low volume hospitals each year.**
*Significance at $P<0.05$.

## Supporting information

**S1 Table. International classification of diseases (ICD) codes used for patient identification and characterization.**
(DOCX)

## Author Contributions

**Conceptualization:** Deanna H. Wong, Joshua G. Cohen.

**Formal analysis:** Alexandra L. Mardock, Yas Sanaiha.

**Methodology:** Deanna H. Wong, Alexandra L. Mardock.

**Resources:** Peyman Benharash.

**Software:** Alexandra L. Mardock, Yas Sanaiha.

**Supervision:** Abdulrahman K. Sinno, Peyman Benharash, Joshua G. Cohen.

**Writing – original draft:** Deanna H. Wong.

**Writing – review & editing:** Deanna H. Wong, Alexandra L. Mardock, Erica N. Manrriquez, Tiffany S. Lai, Peyman Benharash, Joshua G. Cohen.

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
