## [Decision Letter · Decision Letter 0]

6 May 2021

PONE-D-21-00937

Trends in Extent of Surgical Cytoreduction for Patients with Ovarian Cancer

PLOS ONE

Dear Dr. Cohen,

Thank you for submitting your manuscript to PLOS ONE. After careful consideration, we feel that it has merit but does not fully meet PLOS ONE’s publication criteria as it currently stands. Therefore, we invite you to submit a revised version of the manuscript that addresses the points raised during the review process.

Both reviewers found the manuscript interesting and valuable overall. Only minor criticisms were made. I am very confident that once these are incorporated, the manuscript can be published and look forward to the revised version.

We look forward to receiving your revised manuscript.

Kind regards,

Helge Bruns, M.D.

Academic Editor

PLOS ONE

Journal Requirements:

3. Thank you for stating in the text of your manuscript "This study was granted exemption from the Institutional Review Board at the University of California, Los Angeles given its use of deidentified data." Please also add this information to your ethics statement in the online submission form.

4. Thank you for providing the date(s) when patient medical information was initially recorded. Please also include the date(s) on which your research team accessed the databases/records to obtain the retrospective data used in your study.

5. Please state whether there were additional inclusion/exclusion criteria when you selected the included cases.

Reviewers' comments:

Reviewer's Responses to Questions

**Comments to the Author**

1. Is the manuscript technically sound, and do the data support the conclusions?

Reviewer #1: Partly

Reviewer #2: Yes

2. Has the statistical analysis been performed appropriately and rigorously? 

Reviewer #1: Yes

Reviewer #2: Yes

3. Have the authors made all data underlying the findings in their manuscript fully available?

Reviewer #1: Yes

Reviewer #2: Yes

4. Is the manuscript presented in an intelligible fashion and written in standard English?

Reviewer #1: Yes

Reviewer #2: Yes

5. Review Comments to the Author

Reviewer #1: Data from NIS has been analyzed in order to examine the trend of ECR for patients with ovarian cancer. The reviewer has some minor points that need to be addressed:

1. Has there been exclusion criteria? Was invalid, duplicated or missing data detected?

2. In 2015 a transition to ICD 10 took place due to update of a coding system. Has this database change affected the data collected for this research? Has the algorithm for data collection been modified?

3. Although patient characteristics are considered in the study objective, authors have not included this aspekt in their hypothesis.

The authors hypothesize, that ECR is associated with a higher risk of perioperative mortality and length of stay. It is very likely that this effect will occur, when comparing extended surgery procedures against minor ones.

4. Please describe how the reference groups in the multivariate regression were determined.

5. In order to determine if the relative number of ECR procedures has decreased 2013- 2017 it is necessary to indicate the number of new diagnosed ovarian cancer per year. The decrease of the absolute ECR number is most significant since 2016. This coincides with the database change mentioned in item 2 above. Could this have interfered in the results?

6. Which data in this research supports the statement that socioeconomic factors influence the access of surgery? Or is this statement supported by a citation?

7. The increase of neoadjuvant chemotherapy has been observed (cited) between 2004 and 2013. Does recent data support this trend? Moreover the observation in the reviewed research starts at 2013.

8. The decrease in bowel resection could reflect the increasing use of neoadjuvant therapy, but there are also other explanations. As previously mentioned the number of new diagnosed ovarian cancer ist needed in oder to confirm a decrease in bowel resection.

9. How the decrease of ECR relates to the quality of care provided to ovarian cancer patient?

Reviewer #2: I found this manuscript excellent and very interesting to read. It highlights differences in the management of ovarian cancer depending on certain demographic and hospital characteristics.

The only correlation that is difficult to understand is how patients with increased comorbidities had more chances of undergoing extensive surgery for ovarian cancer. The index used in this study is complex with a lot of parameters contributing to the final score and this might account for the finding. If the PS was available it would have made more sense in interpreting the comorbidity index used in this study.

Also I am not familiar with the data base used for extracting data for this study, however its limitations are thoroughly discussed by the author and I don't think these are reducing the value of the results.

I think it should be published at its current form in PLOS ONE

6. PLOS authors have the option to publish the peer review history of their article (what does this mean?). If published, this will include your full peer review and any attached files.

Reviewer #1: **Yes: **Elena Junghans MD

Reviewer #2: **Yes: **Ioannis Biliatis

---

## [Author Response · Author response to Decision Letter 0]

16 Jun 2021

See reviewer response letter. If any further questions, happy to answer.

Thank you,

Joshua Cohen, MD, FACOG, FACS

---

## [Decision Letter · Decision Letter 1]

8 Nov 2021

Trends in Extent of Surgical Cytoreduction for Patients with Ovarian Cancer

PONE-D-21-00937R1

Dear Dr. Joshua Garrett Cohen,

We’re pleased to inform you that your manuscript has been judged scientifically suitable for publication and will be formally accepted for publication once it meets all outstanding technical requirements.

Kind regards,

Xiaodong Cheng

Academic Editor

PLOS ONE

Additional Editor Comments (optional):

All comments have been addressed.

Reviewers' comments:

Reviewer's Responses to Questions

**Comments to the Author**

1. If the authors have adequately addressed your comments raised in a previous round of review and you feel that this manuscript is now acceptable for publication, you may indicate that here to bypass the “Comments to the Author” section, enter your conflict of interest statement in the “Confidential to Editor” section, and submit your "Accept" recommendation.

Reviewer #2: All comments have been addressed

2. Is the manuscript technically sound, and do the data support the conclusions?

Reviewer #2: Yes

3. Has the statistical analysis been performed appropriately and rigorously? 

Reviewer #2: I Don't Know

4. Have the authors made all data underlying the findings in their manuscript fully available?

Reviewer #2: Yes

5. Is the manuscript presented in an intelligible fashion and written in standard English?

Reviewer #2: Yes

6. Review Comments to the Author

Reviewer #2: As I have already described in my initial review of this manuscript I find it excellent and very interesting. I think the authors addressed all issues raised appropriately

7. PLOS authors have the option to publish the peer review history of their article (what does this mean?). If published, this will include your full peer review and any attached files.

Reviewer #2: No

---

## [Editor Report · Acceptance letter]

29 Nov 2021

PONE-D-21-00937R1 

Trends in Extent of Surgical Cytoreduction for Patients with Ovarian Cancer 

Dear Dr. Cohen:

I'm pleased to inform you that your manuscript has been deemed suitable for publication in PLOS ONE. Congratulations! Your manuscript is now with our production department. 

Kind regards, 

on behalf of

Dr. Xiaodong Cheng 

Academic Editor

PLOS ONE